# Efficient Vanadate Removal by Mg-Fe-Ti Layered Double Hydroxide

**Yanwei Guo** [1,2]**, Hongtao Lu** [1]**, Bangjun Han** [1]**, Tiemei Zou** [1] **and Zhiliang Zhu** [2,*]

1   Muncipal and Ecological Engineering School, Shanghai Urban Construction Vocational College, Shanghai 201415, China; guoyanwei@succ.edu.cn (Y.G.); luhongtao@succ.edu.cn (H.L.); hanbangjun@succ.edu.cn (B.H.); zoutiemei@succ.edu.cn (T.Z.)
2   State Key Laboratory of Pollution Control and Resource Reuse, Tongji University, Shanghai 200092, China
*   Correspondence: zzl@tongji.edu.cn; Tel.: +86-21-6598-2426

**Abstract:** A series of novel layered double hydroxides (Mg-Fe-Ti-LDHs) containing $Mg^{2+}$, $Fe^{3+}$ and $Ti^{4+}$ were prepared. The adsorption performance of Mg-Fe-Ti-LDHs on vanadate in aqueous solution was investigated and the effects of various factors on the adsorption process were examined, including initial vanadate concentration, adsorbent dosage, contact time, solution pH and coexisting ions. A preliminary discussion of the adsorption mechanism of vanadate was also presented. Results show that the adsorption efficiency of vanadate increased with the introduction of $Ti^{4+}$ into the laminate of LDHs materials. The adsorption capacity of the materials also differed for different anion intercalated layers, and the Mg-Fe-Ti-LDHs with $Cl^-$ intercalation showed higher vanadate removal compared to the $CO_3^{2-}$ intercalated layer. Furthermore, Mg-Fe-Ti-CLDH showed higher vanadate removal compared to pre-calcination. The adsorption experimental data of vanadate on Mg-Fe-Ti-LDHs were consistent with the Langmuir adsorption isotherm model and the adsorption kinetics followed a pseudo-second order kinetic model. The pH of the solution significantly affected the vanadate removal efficiency. Meanwhile, coexisting ions $PO_4^{3-}$, $SO_4^{2-}$ and $NO_3^-$ exerted a significant influence on vanadate adsorption, the magnitude of the influence was related to the valence state of the coexisting anions. The possible adsorption mechanisms can be attributed to ion exchange and layered ligand exchange processes. The good adsorption capacity of Mg-Fe-Ti-LDHs on vanadate broadens the application area of functional materials of LDHs.

**Keywords:** vanadate removal; titanium; adsorption; layered double hydroxides

## 1. Introduction

Vanadium is abundant in the Earth's crust and is widely used in industry. In the steel industry, vanadium/phosphate systems are used as corrosion inhibitors instead of hexavalent chromium [1]. Large amounts of vanadium are released into the environment and vanadium compounds have been shown to have deleterious effects on the circulatory system, can disrupt human metabolism, and lower dietary glucose, cholesterol and triglycerides in diabetic rats [2–4]. Therefore, vanadium is considered as a potentially dangerous contaminant. It has been included in the list of candidate pollutants by the EPA (Environmental Protection Agency of United States) [5,6]. Removal of vanadium from water bodies has become a major challenge. Vanadium is a redox-sensitive element that occurs in three oxidation states in the environment (+III, +IV and +V), although it usually occurs as the oxygen anion vanadium (V) under most environmental conditions [7,8]. Adsorption is a widely used and very convenient method for removing specific contaminants, especially if the adsorbent has good regenerative properties [9,10]. Extensive work has been carried out to research suitable adsorbents to remove vanadium from aqueous solutions, including chitosan, alumina, anatase, $Fe_2O_3$, LDHs and crystalline hydroxyapatite [11–15].

LDHs have a layered structure with divalent or trivalent cations on the laminate and anions between the layers [16]. The positive charge carried by the laminate can be

neutralized by the interlayer ions, and ion exchange can be realized. Thus, both the inner and outer surfaces of LDHs can provide points for contaminants that are suitable for bonding. Furthermore, LDHs materials are crystalline and have a more stable structure. Benefitting from their higher surface area and anion exchange capacity, LDHs are more advantageous for anion adsorption [17]; they are often studied to treat contaminants such as benzoates, phosphates, cadmium, arsenic, chromates and selenates [18–22].

Related studies have shown that vanadate (V) is readily bound to iron, aluminum and titanium (water) oxides in the form of binary complexes [7]. Fe-based LDHs are functional materials for the effective removal of oxyanions [23,24]. However, studies on the adsorption of vanadate by Fe-based LDHs are still limited. For conventional LDHs materials, the layered materials are mainly divalent and trivalent metal cations. If positive tetravalent metal cations are introduced into the lamellar to replace some of the trivalent cations, the number of positive charges on the lamellar increases and the corresponding number of interlayer anions also increases. According to the ion-exchange properties of LDHs materials, if the number of anions capable of ion-exchange is increased in the appropriate range, the adsorption capacity of the material will be increased. The proposal of this research was to synthesize a series of novel materials (Mg-Fe-Ti-LDHs) and use these to study the adsorption properties of vanadate in water. The factors affecting the adsorption process—such as pH, reaction time, initial pollutant concentration and coexisting ions—were experimentally evaluated, and the adsorption isotherms as well as the adsorption kinetics were fitted. Furthermore, the possible adsorption mechanisms of vanadate on the synthesized material were discussed. This study is indicative for the study of vanadate removal in water bodies.

## 2. Materials and Methods

### 2.1. Materials and Synthesis

$MgCl_2 \cdot 6H_2O$, $FeCl_3 \cdot 6H_2O$, $TiCl_4$, HCl, NaOH and $Na_2CO_3$ were used in this research. The vanadate stock solution was obtained by dissolving the sodium metavanadate in deionized water. All original reagents are analytically pure.

Co-precipitation method is used to synthesize related materials, including Mg-Fe-Ti-LDHs, with carbonate and chloride ion intercalations. For the carbonate intercalation, a mixed solution containing $MgCl_2 \cdot 6H_2O$, $FeCl_3 \cdot 6H_2O$ and $TiCl_4$ was prepared with deionized water (concentration is 0.6 mol/L, 0.196 mol/L and 0.004 mol/L), in which $Mg^{2+}/(Fe^{3+}+Ti^{4+})$ ratio is 3. Another solution containing NaOH (4.8 mol/L) and $Na_2CO_3$ (1.2 mol/L) was prepared with deionized water. The above solutions were added simultaneously and slowly dropped into vigorously stirred deionized water. The whole reaction mixture was aged at 353 K for 20 h and then the precipitate obtained from the reaction was repeatedly washed with deionized water until there were no residual reactants on the surface of the precipitate. Finally, the resulting solid was dried at 423 K and ground into powders and the obtained LDHs material was recorded as $Mg$-$Fe$-$Ti$-$LDH$-$CO_3{}^{2-}$. The $Mg$-$Fe$-$Ti$-$LDH$-$CO_3{}^{2-}$ was calcined in a muffle furnace at 773 K to get the calcined Mg-Fe-Ti-LDHs (named Mg-Fe-Ti-CLDH).

As for the Mg-Fe-Ti-LDHs with chloridion intercalation, it was synthesized using a method similar to the one above. First, a mixed solution containing $MgCl_2 \cdot 6H_2O$ (1.8 mol/L), $FeCl_3 \cdot 6H_2O$ (*x* mol/L) and $TiCl_4$ (*y* mol/L) (*x* + *y* = 0.6 mol/L, *y* = 0, 0.012, 0.03, 0.06, 0.12 or 0.3 mol/L) with $Mg^{2+}/(Fe^{3+} + Ti^{4+})$ ratio of 3 was made. Another solution of NaOH (4.8 mol/L) was made in 50 mL deionized water. The same slow titration co-precipitation reaction process was used, but with nitrogen protection throughout the reaction to prevent $CO_2$ in air influence. The resulting product was first aged at 353 K for 20 h. The precipitate obtained from the reaction was repeatedly washed with deionized water until there were no residual reactants on the surface of the precipitate. Finally, the resulting solid was dried at 423 K and ground into powders. The obtained materials were named Mg-Fe-Ti-LDH-Cl⁻-1 (y = 0 mol/L), Mg-Fe-Ti-LDH-Cl⁻-2 (y = 0.012 mol/L), Mg-Fe-Ti-

LDH-Cl$^-$-3 (y = 0.03 mol/L), Mg-Fe-Ti-LDH-Cl$^-$-4 (y = 0.06 mol/L), Mg-Fe-Ti-LDH-Cl$^-$-5 (y = 0.012 mol/L) and Mg-Fe-Ti-LDH-Cl$^-$-6 (y = 0.3 mol/L).

### 2.2. Material Characterization and Vanadate Analysis

A Bruker Diffractometer (D8 Advance, Bruker-AXS, Karlsruhe, Germany) was used to measure the X-ray diffraction pattern of the Mg-Fe-Ti-LDHs (CuK$\alpha$ radiation operated at 40 kV and 40 mA). A FTIR instrument (Thermo Nicolet 5700, Thermo Nicolet Corporation, Waltham, MA, USA ) was used to record the infrared spectra (FTIR) of the Mg-Fe-Ti-LDHs. A simultaneous thermal analyzer (SDT Q600, TA Instruments Inc., New Castle, DE, USA) was used to research the thermal characterization of the LDHs. A scanning electron microscope (Quanta 200 FEG, EFI Company, Amsterdam, Holland) was used to observe the fine structure of the material. The vanadate concentration in the aqueous solution was determined by ICP-OES (Optima2100, Perkin-Elmer company, Waltham, MA, USA).

### 2.3. Adsorption Experiments

Experiments related to adsorption studies were carried out in batches using a shaking incubator at 298 K and 150 rpm. The suspension was filtrated and then analyzed for residual vanadate concentration. In adsorptive isothermal experiments, vanadate concentrations ranged from 1 to 20 mg/L, the initial pH of the solution was set to 5.0 using HCl and NaOH solutions, and during the adsorption process, the solution pH was not adjusted. A total of 10 mg of Mg-Fe-Ti-LDHs was taken into 50 mL of vanadate solution and sustained 24 h in order to reach equilibriumed by membrane. In the time effect experiment, the time intervals were set at 10. After reaching equilibrium, the LDHs were filter min, 15 min, 30 min, 60 min, 120 min, 240 min, 360 min and 480 min, respectively. The effect of the initial solution pH on the adsorption of vanadate was evaluated using 10 mg/L vanadate solutions at an LDHs concentration of 0.2 g/L, and the pH of the starting solutions were set 4, 5, 6, 7, 8, 9, 10, 11 and 12, respectively. For the purpose of researching the adsorbent dosage on vanadate removal, the Mg-Fe-Ti-LDHs dosages were set 5 mg, 10 mg, 25 mg, 50 mg and 75 mg, with the initial vanadate of 10 mg/L; the solution pH was 5.0. Effects of coexistent anions ($NO_3^-$, $SO_4^{2-}$ and $PO_4^{3-}$) on vanadate adsorption were investigated by adding 10 mg of the Mg/Fe/Ti-LDHs to 50 mL of vanadate solutions containing a certain concentration of anions (pH 5.0), in which initial vanadate concentration was 5 mg/L and competing anions was 50 mg/L, respectively. These were shaken at 150 rpm, 298 K. After 24 h, the aqueous samples were filtered and analyzed. All the adsorption experiments were conducted in triplicate and the mean values were reported.

The amount of adsorption $q_t$ was attained by the following equation [25]:

$$q_t = \frac{(C_0 - C_t)V}{m} \tag{1}$$

$C_0$ (mg/L): initial vanadate concentration; $C_t$ (mg/L): the concentration corresponding to a reaction time of ($t$) in solution; $m$ (mg): the mass of the sorbent; $V$ (L): the volume of solution.

### 2.4. Desorption Experiment

Desorption experiment was operated to evaluate the adsorbent reusability. A total of 20 mg of Mg-Fe-Ti-LDHs was added to 50 mL vanadate solutions ($C_0$: 10 mg/L) and shaken in a incubator at 298 K and 150 rpm. After equilibrium (24 h), the suspension was filtered and the supernatant was used to analyze vanadate concentration. The material was collected after analysis and dried at 100 °C. Ten milligrams of the dried material was taken into 50 mL $Na_3PO_4$ solution and shaken at 150 rpm and 298 K ($C_0$: 500 mg/L). After 24 h, the supernatant was taken and the concentration of the remaining vanadate was measured.

## 3. Results

### 3.1. Characterization

3.1.1. XRD Analysis

The X-ray diffraction pattern of the LDHs are shown in Figure 1. We can observe that Mg-Fe-Ti-LDHs-Cl$^-$ and Mg-Fe-Ti-LDH-CO$_3$$^{2-}$ show a series of sharp and symmetric diffraction and broad asymmetric peaks of LDHs, which were consistent with other studies [26]. Further increases of titanium content led to a decrease of the phase crystallization of the samples, such as Mg-Fe-Ti-LDH-Cl$^-$-5 and Mg-Fe-Ti-LDH-Cl$^-$-6. Although the main characteristic of LDHs can still be found in Mg-Fe-Ti-LDH-Cl$^-$-5, byproducts had been produced. Furthermore, weak diffraction peaks of MgO phase were found in the XRD patterns of the Mg-Fe-Ti-CLDH.

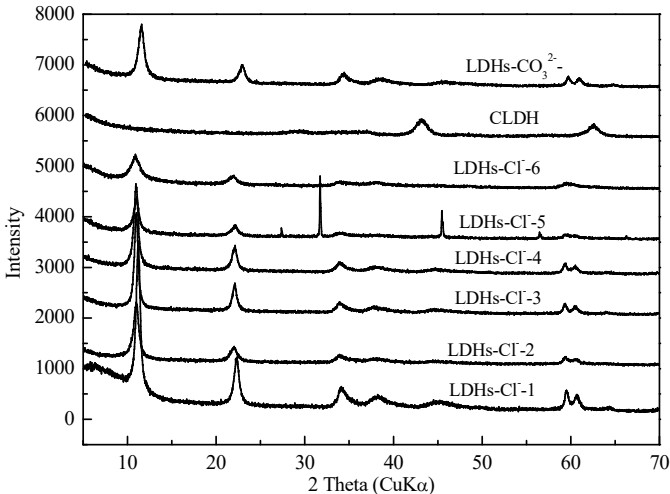

**Figure 1.** XRD patterns of Mg-Fe-Ti-LDHs and Mg-Fe-Ti-CLDH.

3.1.2. Thermo Analysis

As shown in Figure 2, the TG-DTA results for Mg-Fe-Ti-LDH-CO$_3$$^{2-}$ and Mg-Fe-Ti-LDH-Cl$^-$-2 have the typical characteristics of LDHs [27]. For Mg-Fe-Ti-LDH-CO$_3$$^{2-}$, two significant weight loss inflection points occur; the first one is around 200 °C, which is mainly the removal of H$_2$O from the surface and interlayer spaces, with a weight reduction of about 14.5%. The second one is around 400 °C, which mainly refers to the loss of interlayer CO$_3$$^{2-}$ and de-hydroxylation of the layer, with a weight reduction of about 23.2%. The Mg-Fe-Ti-LDHs-Cl$^-$-2 showed a distinctly different tendency, having three different weight reduction temperature intervals. The first one is at a temperature around 150 °C, which was mainly the removal of H$_2$O from the surface and the interlayer, and the weight loss was about 13.1%. The second and third temperatures were about 300 °C and 350 °C, respectively, which is mainly the de-hydroxylation and chloridion removal process. The weight loss during the second and third process was 22.2%. As for the Mg-Fe-Ti-CLDH material, two weight losses could be seen at about 350 °C and 530 °C, resulting in the removal of the surface water and CO$_2$.

3.1.3. FTIR

The FTIR spectra of Mg-Fe-Ti-LDHs-Cl$^-$-2, Mg-Fe-Ti-LDHs-CO$_3$$^{2-}$ and Mg-Fe-Ti-CLDH materials is shown in Figure 3, and the infrared spectra of Mg-Fe-Ti-LDHs materials conform to the FTIR spectra of typical layered structure materials [28,29]. For LDHs materials intercalated with carbonate ions, the -OH hydrogen-oxygen bond stretching vibration peak appears at 3587 cm$^{-1}$. The bending vibration peak of crystal water -OH appears around 1653 cm$^{-1}$ due to the adsorption of water on the surface of hydrargyrite and the insertion of some H$_2$O in the interlayer. The spectral peak of the Mg-Fe-Ti-LDHs sample near 1363 cm$^{-1}$ is caused by the asymmetric stretching vibration of C-O in CO$_3$$^{2-}$.

The peaks appearing in the region from 500 to 1000 cm$^{-1}$ are peaks of metal-oxygen and metal-hydrogen-oxygen bonds [30]. Figure 3 shows that the intensity of each peak in the infrared spectrum of the Mg-Fe-Ti-CLDH material is significantly reduced, especially the peak of $CO_3^{2-}$. Mainly due to high calcination temperature, the carbonate radicals are emitted in the form of carbon dioxide and the material structure is destroyed. For the Mg-Fe-Ti-LDHs-Cl$^{-}$-2 material, the peaks at 3445 cm$^{-1}$ and 1635 cm$^{-1}$ are also the vibration peaks of water molecules and the bending vibration peaks of -OH. The C-O asymmetric stretching vibration of $CO_3^{2-}$ is the reason for the formation of the peak at 1365 cm$^{-1}$. During the synthesis of the material, some of the $CO_2$ in the air enters the reaction system and forms carbonate.

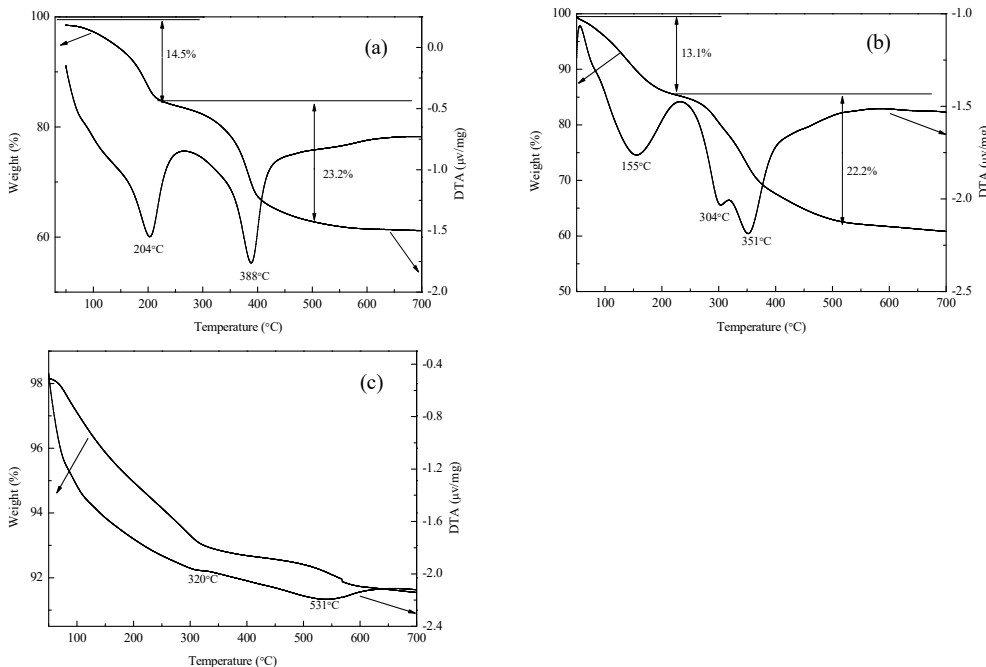

**Figure 2.** (**a**) TG-DTA curves of Mg-Fe-Ti-LDHs-$CO_3^{2-}$, (**b**) Mg-Fe-Ti-LDHs-Cl$^{-}$-2 and (**c**) Mg-Fe-Ti-CLDH.

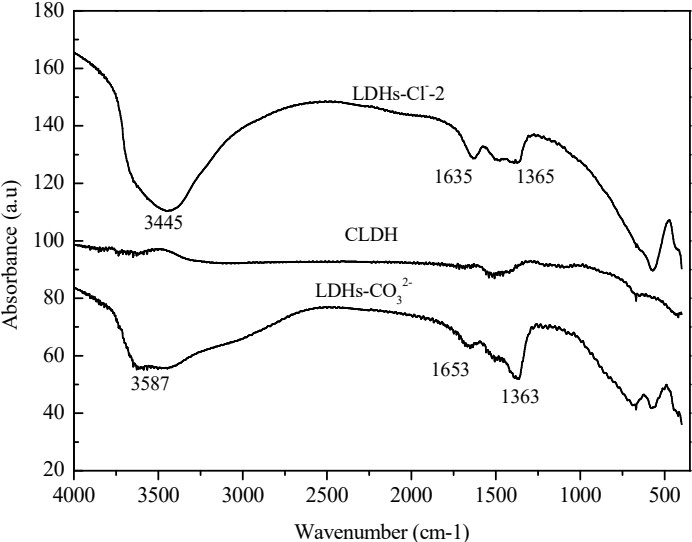

**Figure 3.** FTIR patterns of Mg-Fe-Ti-LDHs.

### 3.2. Comparison of Vanadate Removal by Different Kinds of Mg-Fe-Ti-LDHs

The removal efficiency of vanadate by different kinds of Mg-Fe-Ti-LDHs is shown in Figure 4. Each adsorbent has a distinctively different removal efficacy on the vanadate. The adsorption efficiency of the vanadate was found to be Mg-Fe-Ti-LDHs-Cl$^-$-2 > Mg-Fe-Ti-LDHs-Cl$^-$-1 > Mg-Fe-Ti-CLDH > Mg-Fe-Ti-LDHs-CO$_3^{2-}$ in order, indicating that the interaction of LDHs with chloride ions was more efficient for vanadate removal than that with carbonates. With the increasing concentration of Ti$^{4+}$ in the layers, the vanadate removal efficiency increased. Meanwhile, the Mg-Fe-Ti-CLDH which was obtained from the calcination of the Mg-Fe-Ti-LDHs-CO$_3^{2-}$ showed a greater vanadate removal effect.

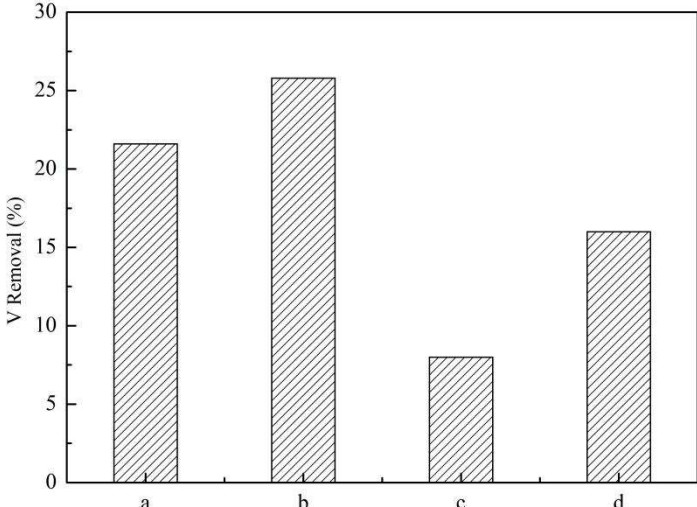

**Figure 4.** Removal of vanadate by (a) Mg-Fe-Ti-LDHs-Cl-1, (b)Mg-Fe-Ti-LDHs-Cl-2, (c) Mg-Fe-Ti-LDHs-CO$_3^{2-}$ and (d) Mg-Fe-Ti-CLDH (experimental conditions: $C_0$ = 50 mg/L, adsorbent dosage = 10 mg, pH = 5.0, contact time = 24 h, T = 298 K).

### 3.3. Influence of pH Value

Figure 5 shows that the vanadate adsorbed per unit dosage of LDHs decreases as the pH of the solution increases. The highest value of adsorption was reached at pH 4.0 and the lowest was at pH 12.0. The reason for the pH effect on vanadate adsorption could be explained by the fact that when the pH of the solution is low, the hydrogen ion content of the solution increases, making the surface of the adsorbent positively charged and strengthening the attraction of the vanadate anion to the surface of the adsorbent. As the solution pH increases, the hydroxyl ion becomes dominant and competes with the vanadate anion for adsorption.

### 3.4. Adsorbent Dosage

Figure 6 reflects the effect of adsorbent dosage on pollutant removal. It shows that the vanadate adsorption efficiency increases rapidly with increasing adsorbent dosage. The curve leveled off when the LDHs dosage exceeded 1.0 g/L. When the Mg-Fe-Ti-LDHs-Cl$^-$-2 and Mg-Fe-Ti-CLDH reached 1.5 g/L, the removal efficiency of vanadate was almost 100%.

### 3.5. Adsorption Isotherm Studies

Figure 7 represents the adsorption isotherms of vanadate onto the synthesized Mg/Fe/Ti-LDHs.

Results show that for the three materials, Mg-Fe-Ti-LDHs-Cl$^-$-2, Mg-Fe-Ti-CLDH and Mg-Fe-Ti-LDHs-CO$_3^{2-}$, the adsorption capacity increases rapidly with the increase of vanadate equilibrium concentration until the maximum adsorption is reached. The adsorption capacities of the three LDHs materials were, in order, Mg-Fe-Ti-LDHs-Cl$^-$-2 > Mg-Fe-Ti-CLDH > Mg-Fe-Ti-LDHs-CO$_3^{2-}$.

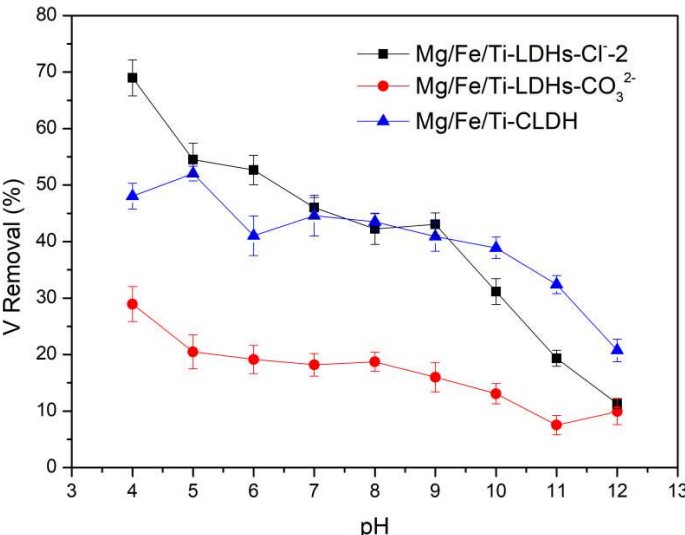

**Figure 5.** Effects of initial solution pH on vanadate adsorption onto Mg-Fe-Ti-LDHs-Cl$^-$-2, Mg-Fe-Ti-LDHs-CO$_3$$^{2-}$ and Mg-Fe-Ti-CLDH (experimental conditions: $C_0$ = 10 mg/L, adsorbent dosage = 10 mg, pH = 4.0–12.0, contact time = 24 h, T = 298 K).

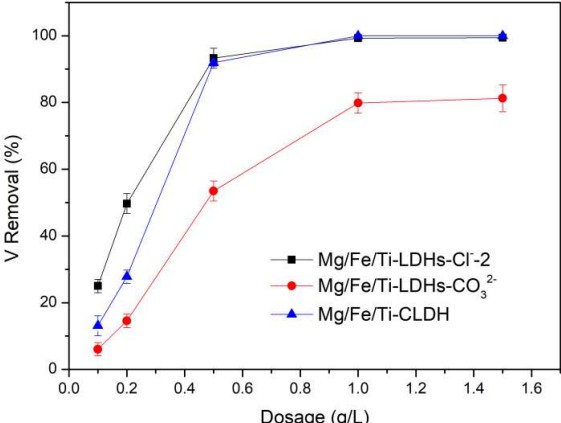

**Figure 6.** Adsorbent dosage on vanadate adsorption onto Mg-Fe-Ti-LDHs-Cl$^-$-2, Mg-Fe-Ti-LDHs-CO$_3$$^{2-}$ and Mg-Fe-Ti-CLDH (experimental conditions: $C_0$ = 10 mg/L, adsorbent dosage = 5–75 mg, pH = 5, contact time = 24 h, T = 298 K).

The isotherm models of Langmuir and Freundlich are applied to analyze the experimental adsorption data. The general formula of Langmuir model is [31]:

$$\frac{C_e}{q_e} = \frac{C_e}{q_m} + \frac{1}{K_L q_m} \tag{2}$$

The general formula of Freundlich model is [29]:

$$\ln q_e = \ln K_F + \frac{1}{n} \ln C_e \tag{3}$$

$C_e$ (mg/L): vanadate equilibrium concentration; $q_e$ (mg/g): vanadate equilibrium adsorption capacity; $K_L$ (L/mg): the Langmuir adsorption constant; $q_m$ (mg/g): the saturated adsorption capacity; $1/n$ is the heterogeneity factor and $K_F$ (L/mg) is the Freundlich constant.

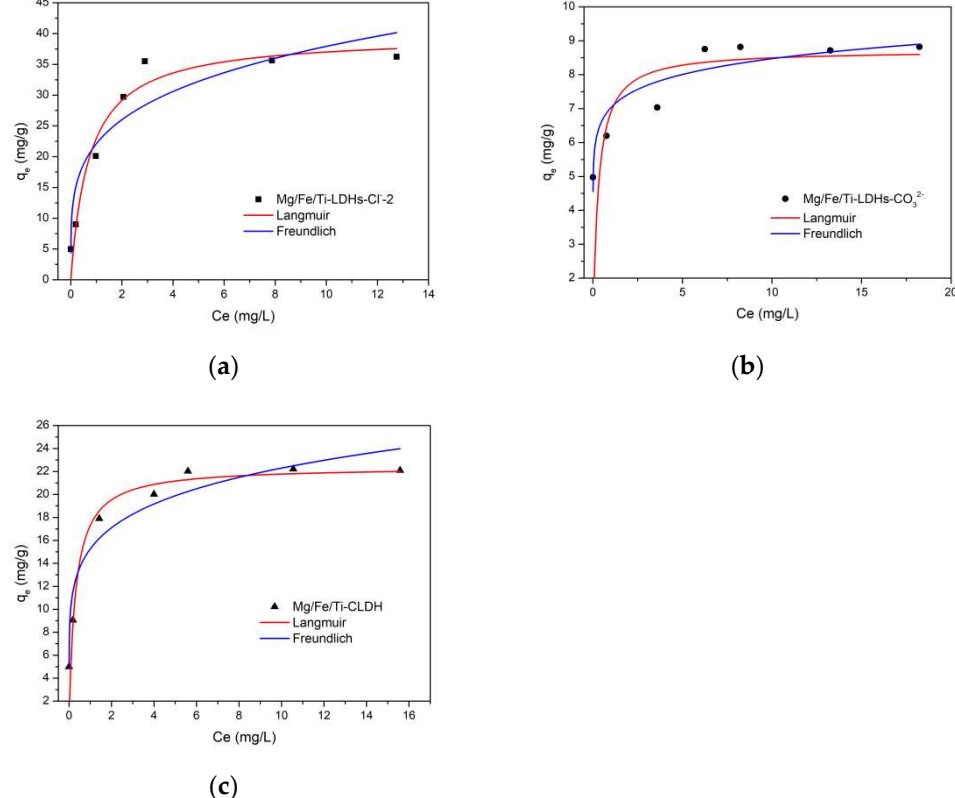

**Figure 7.** Adsorption isotherms of vanadate on the synthesized Mg-Fe-Ti-LDHs-Cl⁻-2 (**a**), Mg-Fe-Ti-LDHs-$CO_3^{2-}$ (**b**) and Mg-Fe-Ti-CLDH (**c**) (experimental conditions: $C_0$ = 1–20 mg/L, adsorbent dosage = 10 mg, pH = 5.0, contact time = 24 h, T = 298 K).

Table 1 shows the results of the Langmuir and Freundlich isotherm models for vanadate removal from Mg-Fe-Ti-LDHs-Cl⁻-2, Mg-Fe-Ti-LDHs-$CO_3^{2-}$ and Mg-Fe-Ti-CLDH. The adsorption process of vanadate on LDHs materials is more consistent with the Langmuir isotherm model, and therefore vanadate adsorption on these LDHs is similar to monolayer adsorption. According to the formula of Langmuir isotherm, the saturated adsorption capacity of vanadate on the LDHs was 38.5 mg/g, 9.01 mg/g and 22.8 mg/g, respectively. Table 2 showed the comparison results of the maximum adsorption capacities of various adsorbents for vanadate. It is found that the LDHs in this work have a relatively higher adsorption capacity, which means they can be used as a potential efficient adsorbent for the vanadate from aqueous solutions.

**Table 1.** Adsorption isotherm models of $VO_3^-$ adsorption onto Mg-Fe-Ti-LDHs-Cl⁻-2, Mg-Fe-Ti-LDHs-$CO_3^{2-}$ and Mg-Fe-Ti-CLDH.

| LDHs Material | Langmuir Isotherm | | | Freundlich Isotherm | | |
|---|---|---|---|---|---|---|
| | $q_m$ | $K_L$ | $R^2$ | $K_F$ | $1/n$ | $R^2$ |
| Mg-Fe-Ti-LDHs-Cl⁻-2 | 38.5 | 2.60 | 0.996 | 23.4 | 0.233 | 0.903 |
| Mg-Fe-Ti-LDHs-$CO_3^{2-}$ | 9.01 | 3.00 | 0.997 | 7.06 | 0.074 | 0.887 |
| Mg-Fe-Ti-CLDH | 22.8 | 4.40 | 0.999 | 15.1 | 0.168 | 0.958 |

### 3.6. Kinetic Studies

In order to study the time effect on vanadate adsorption onto Mg-Fe-Ti-LDHs-Cl⁻-2, Mg-Fe-Ti-LDHs-$CO_3^{2-}$ and Mg-Fe-Ti-CLDH, the adsorption capacity variations with time were investigated. The effect of contact time on vanadate adsorption onto Mg/Fe/Ti-LDHs is shown in Figure 8. It is found that the adsorption rate was fast at the first 100 mins because the adsorption sites were more abundant and available at the beginning. As the

adsorption sites were gradually filled with the vanadate, the adsorption process slowed down and gradually reached the equilibrium.

**Table 2.** Comparison of several reported adsorbents for vanadate.

| Adsorbent | $q_m$ | Reference |
|---|---|---|
| Mg-Fe-Ti-LDHs-Cl$^-$-2 | 38.5 (mg/g) | This study |
| Mg-Fe-Ti-LDHs-CO$_3$$^{2-}$ | 9.01 (mg/g) | This study |
| Mg-Fe-Ti-CLDH | 22.8 (mg/g) | This study |
| γ-AlOOH | 3.61(mmol/g) | [32] |
| Silica | 81.0 (mg/kg) | [33] |
| HA + silica | 166.7 (mg/kg) | [34] |
| Fe(III)/Cr(III) hydroxide | 11.43 (mg/g) | [35] |

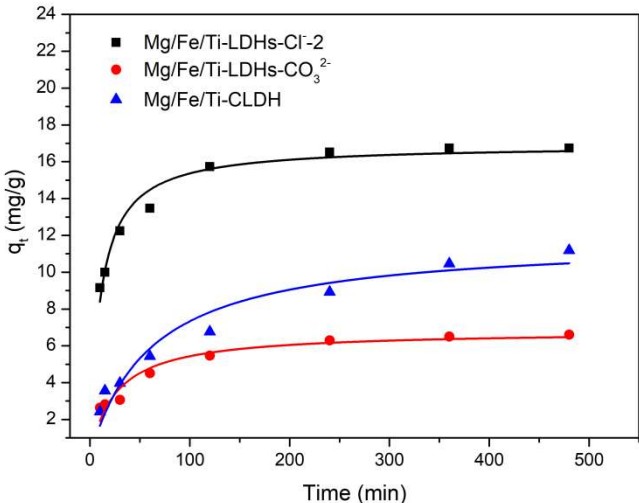

**Figure 8.** Effect of contact time on vanadate adsorption onto Mg-Fe-Ti-LDHs-Cl$^-$-2, Mg-Fe-Ti-LDHs-CO$_3$$^{2-}$ and Mg-Fe-Ti-CLDH (experimental conditions: $C_0$ = 5 mg/L, adsorbent dosage = 10 mg, pH = 5.0, contact time = 10–480 min, T = 298 K).

The adsorption kinetics are identified by a pseudo-first order kinetic model and a pseudo-second order kinetic model.

The general formula of pseudo-first order kinetic model is [30]:

$$\ln(q_e - q_t) = \ln q_e - k_1 t \tag{4}$$

The general formula of pseudo-second order kinetic model is [36]:

$$\frac{t}{q_t} = \frac{1}{k_2 q_e^2} + \frac{t}{q_e} \tag{5}$$

$q_t$ (mg/g): vanadate adsorbed at time $t$; $q_e$ (mg/g): the vanadate adsorbed at equilibrium time; $k_1$ (min$^{-1}$): the rate constant of pseudo-first order; $k_2$ (g/(mg·min)): the rate constant of pseudo second-order.

Table 3 shows the kinetic parameters for sorption of vanadate by Mg-Fe-Ti-LDHs-Cl$^-$-2, Mg-Fe-Ti-LDHs-CO$_3$$^{2-}$ and Mg-Fe-Ti-CLDH. It can be seen that the pseudo second-order model was more consistent with the experimental results. This result indicates the process of pollutant adsorption is still dominated by the occupation of active sites.

*3.7. Thermodynamics Studies*

Figure 9 shows the adsorption efficiency of vanadate onto Mg-Fe-Ti-CLDH at different temperatures. The vanadate equilibrium adsorption capacity on Mg-Fe-Ti-CLDH increases with the increasing reaction temperature, which proves that the interaction between

vanadate and Mg-Fe-Ti-CLDH is endothermic. The general formula of thermodynamic parameters $\Delta G°$ (kJ/mol), $\Delta H°$ (kJ/mol) and $\Delta S°$ (kJ/mol·K) are [37,38]:

$$\Delta G° = -RT \ln K_L \tag{6}$$

$$\ln K_L = \frac{\Delta S°}{R} - \frac{\Delta H°}{RT} \tag{7}$$

**Table 3.** Adsorption kinetic models for vanadate adsorption by Mg-Fe-Ti-LDHs-Cl$^-$-2, Mg-Fe-Ti-LDHs-CO$_3{}^{2-}$ and Mg-Fe-Ti-CLDH.

| LDHs Material | Pseudo-First Order Kinetic | | | Pseudo-Second Order Kinetic | | |
|---|---|---|---|---|---|---|
| | $k_1$ | $q_{e,cal}$ | $R^2$ | $k_2$ | $q_e$ | $R^2$ |
| LDHs-Cl$^-$-2 | 0.0021 | 11.0 | 0.954 | 0.005 | 17.5 | 0.999 |
| LDHs-CO$_3{}^{2-}$ | 0.01 | 4.33 | 0.996 | 0.005 | 6.99 | 0.998 |
| Mg-Fe-Ti-CLDH | 0.006 | 9.05 | 0.983 | 0.001 | 12.2 | 0.984 |

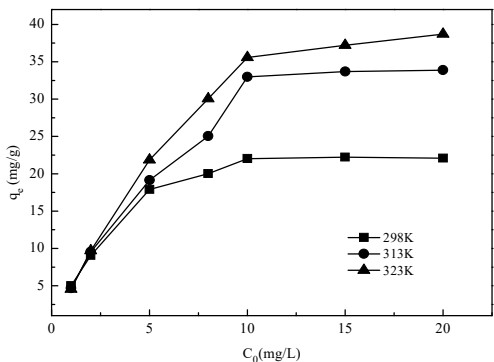

**Figure 9.** Effect of temperature on vanadate adsorption onto Mg-Fe-Ti-CLDH (experimental conditions: $C_0$ = 1–20 mg/L, adsorbent dosage = 10 mg, pH = 5.0, contact time = 24 h, T = 298–323 K).

*R:* the gas constant 8.314 J/mol·K; $K_L$ (L/mol): the Langmuir constant; *T* (K): the absolute temperature.

Figure 10 shows the relationship between lnKc$^0$ and 1/T, and the calculated thermodynamic parameters for vanadate adsorption on Mg-Fe-Ti-CLDH are given in Table 4. The results show that $\Delta G°$ is negative, indicating the removal of vanadate by Mg-Fe-Ti-CLDH is spontaneous. $\Delta H°$ is positive, confirming that the vanadate adsorption process is endothermic, and $\Delta S°$ is positive, showing the increment in stoichiometry at the solid and solution interface which is caused by structural changes in the vanadate-Mg-Fe-Ti-CLDH system during the adsorption process.

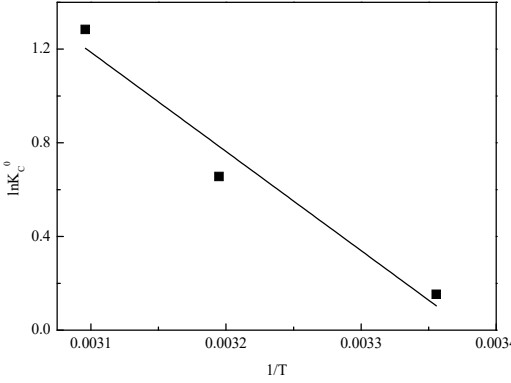

**Figure 10.** Plot of lnKc$^0$ versus 1/T on adsorption of VO$_3{}^-$ onto Mg-Fe-Ti-CLDH.

**Table 4.** Adsorption thermodynamic for vanadate removal from Mg-Fe-Ti-CLDH.

| $\Delta H°$ | $\Delta S°$ | $\Delta G°$ | | | $R^2$ |
| --- | --- | --- | --- | --- | --- |
| | | **298 K** | **313 K** | **323 K** | |
| 35.2 | 0.119 | −0.378 | −1.71 | −3.45 | 0.960 |

*3.8. Adsorption Mechanism*

Figure 11 shows the XRD patterns of the LDHs material after vanadate adsorption. For Mg-Fe-Ti-LDHs-Cl$^-$-2 and Mg-Fe-Ti-LDHs-CO$_3{}^{2-}$, the XRD patterns after vanadate adsorption still show the typical crystalline peaks, which indicates that the layered structure is preserved, proving that the adsorption is an ion exchange process and vanadate replaces the interlayer anion and enters the interlayer. According to the XRD pattern, the metal–metal distance and interlayer distance of the Mg-Fe-Ti-LDHs-Cl$^-$-2 and Mg-Fe-Ti-LDHs-CO$_3{}^{2-}$ before and after adsorption of vanadate can be calculated, and the results are shown in Table 5. These results show that after adsorption of vanadate, the metal ion spacing of the two materials did not change, while the layer spacing changed significantly; for the Mg-Fe-Ti-LDHs-Cl$^-$-2 material, the interlayer distance changed from 24.2 Å before adsorption to 24.0 Å. This indicates that the adsorption process of vanadate for both materials is an ion exchange process in which the anions in the interlayer exchange with vanadate ions in solution, causing vanadate ions in solution to enter the interlayer.

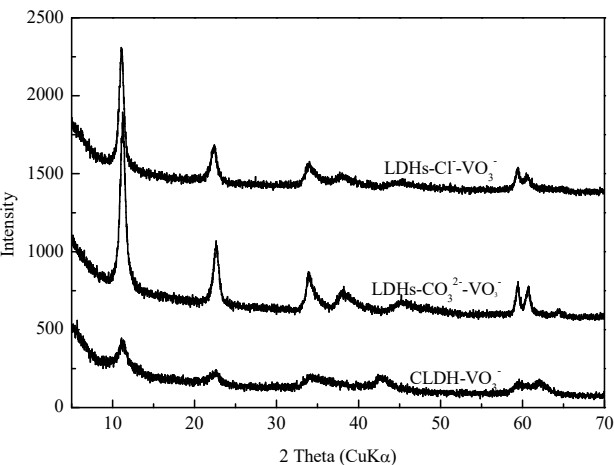

**Figure 11.** XRD pattern of Mg-Fe-Ti-LDHs-Cl$^-$-2, Mg-Fe-Ti-LDHs-CO$_3{}^{2-}$ and Mg-Fe-Ti-CLDH after vanadate adsorption.

**Table 5.** Crystal parameters of Mg/Fe/Ti-LDHs-Cl$^-$-2, Mg/Fe/Ti-LDHs-CO$_3{}^{2-}$ and Mg/Fe/Ti-CLDH after adsorption of VO$_3{}^-$.

| Material | $d$ (003) (Å) | Metal-Metal Distance ($a$) (Å) | Interlayer Distance ($c$) (Å) |
| --- | --- | --- | --- |
| Mg-Fe-Ti-LDHs-Cl$^-$-2 before adsorption | 8.06 | 3.11 | 24.2 |
| Mg-Fe-Ti-LDHs-Cl$^-$-2 after adsorption | 7.99 | 3.11 | 24.0 |
| Mg-Fe-Ti-LDHs-CO$_3{}^{2-}$ before adsorption | 7.81 | 3.11 | 23.4 |
| Mg-Fe-Ti-LDHs-CO$_3{}^{2-}$ after adsorption | 7.87 | 3.11 | 23.6 |
| CLDH after adsorption | 7.86 | 3.11 | 23.6 |

For Mg-Fe-Ti-CLDH, the laminar structure is reconstructed after vanadate adsorption. Therefore, the possible adsorption mechanism is that the vanadate anions were inserted into the interlayer of LDHs by chemisorption, making Mg-Fe-Ti-CLDH reconstruct the lamellar structure.

Figure 12 shows the SEM images of Mg-Fe-Ti-LDHs before and after vanadate adsorption. It can be seen that the Mg-Fe-Ti-LDHs-Cl$^-$-2 after vanadate adsorption shows a clear layered morphology. This indicates that vanadate adsorption on Mg-Fe-Ti-LDHs-Cl$^-$-2 did not change the initial structure, and confirms that the pathway of vanadate adsorption onto the LDHs is ion-exchange. This result is the same as other previous studies on LDHs [39]. For Mg-Fe-Ti-CLDH, before vanadate adsorption, SEM images showed no layered structure. Meanwhile, after vanadate adsorption, SEM images showed a clear layered structure, indicating also that the mechanism of vanadate adsorption on Mg-Fe-Ti-CLDH is structural reconstruction.

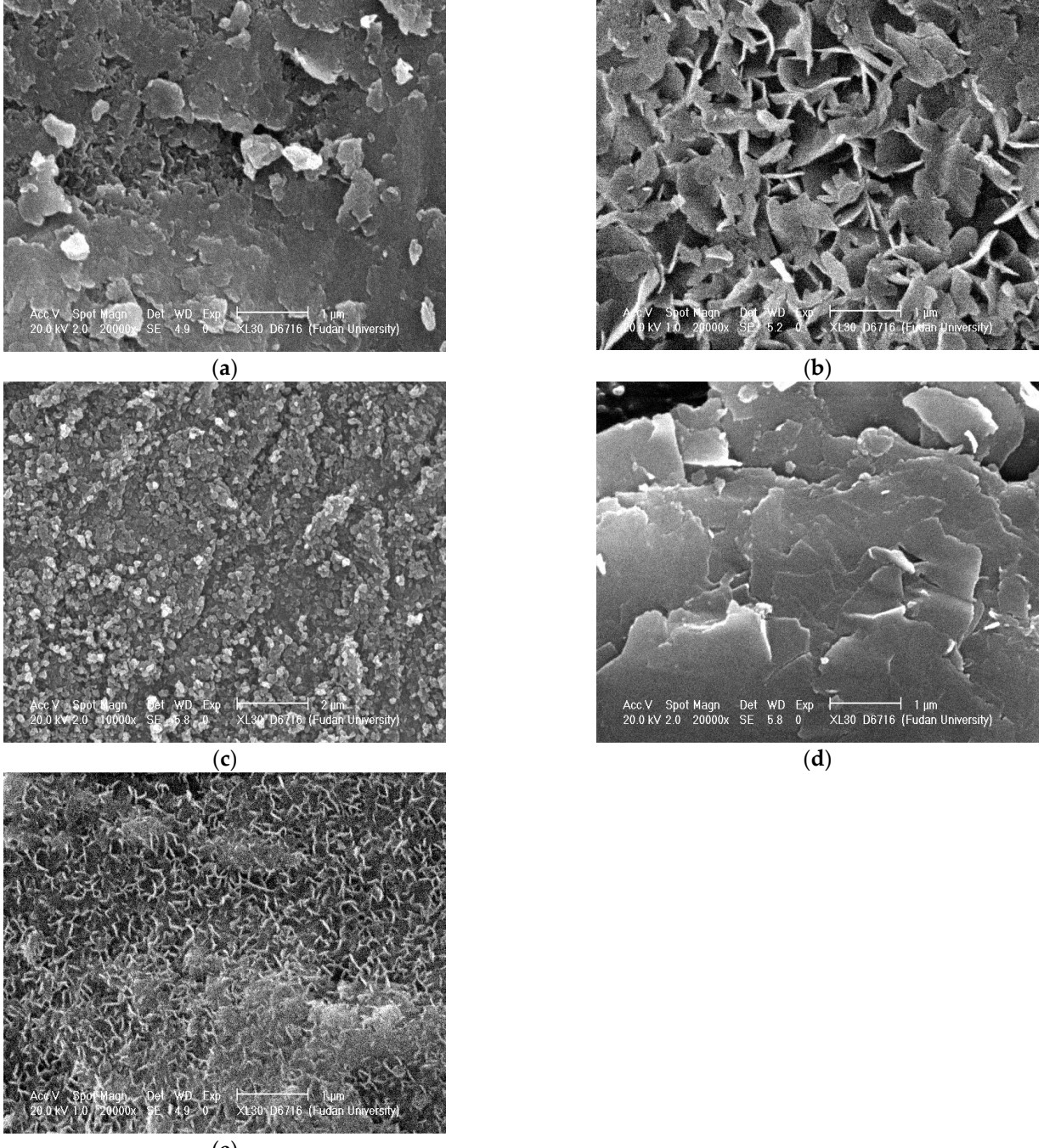

**Figure 12.** The SEM images of Mg-Fe-Ti-LDHs before (**a**) Mg-Fe-Ti-LDHs-Cl$^-$-2, (**b**) Mg-Fe-Ti-LDHs-CO$_3{}^{2-}$, (**c**) Mg-Fe-Ti-CLDH) and after (**d**) Mg-Fe-Ti-LDHs-Cl$^-$-2, (**e**) Mg-Fe-Ti-CLDH) vanadate adsorption.

### 3.9. Effect of Coexistent Ions

The background salts in polluted water may have an effect on the adsorption process, since there is a relatively low concentration of the background salts in common simulated adsorption systems. This effect was further discussed in the section on coexistent ions effect in our study. Natural waters often contain various kinds of ions as background salts, which can affect the vanadate adsorption. In real systems such as waste water treatment, humic and fulvic acids also interfere with the adsorption of oxyanions [40]. Therefore, the effect of coexistent ions on the removal of vanadate by the synthesized Mg-Fe-Ti-LDHs has been examined. This includes the effects of three different coexisting anions, $NO_3^-$, $SO_4^{2-}$ and $PO_4^{3-}$, on the uptake of vanadate using a 1:10 ration of vanadate to competing anions in aqueous solution.

Table 6 indicates that the adsorption of vanadate onto Mg-Fe-Ti-LDHs is mainly influenced by the valence state of the competing anions. The higher the valence state of the anions, the greater the effect on vanadate. Thus, the effect of competing anions on vanadate adsorption is $PO_4^{3-} > SO_4^{2-} > NO_3^-$. This result is the same as that observed in a previous report [41]. The possible reason for the result is that the affinity of LDHs materials for anions increases with increasing charge density of anions [42,43].

**Table 6.** Effect of coexisting ions on Mg-Fe-Ti-LDHs-$Cl^-$-2, Mg-Fe-Ti-LDHs-$CO_3^{2-}$ and Mg-Fe-Ti-CLDH adsorption of $VO_3^-$.

| Materials | Vanadate Removal (%) | | | |
|---|---|---|---|---|
| | No Coexisting Ions | $NO_3^-$ | $SO_4^{2-}$ | $PO_4^{3-}$ |
| LDHs-$Cl^-$-2 | 80.4 | 67.6 | 60.7 | 45.1 |
| LDHs-$CO_3^{2-}$ | 28.1 | 26.1 | 24.5 | 10.7 |
| Mg-Fe-Ti-CLDH | 71.6 | 68.6 | 58.5 | 10.2 |

### 3.10. Desorption Process

Desorption performance is a prominent indicator by which to evaluate the reusability of an adsorbent. Therefore, desorption experiments of vanadate on Mg-Fe-Ti-LDHs were carried out. Table 6 shows that the effect of various competing anions on the adsorption of vanadate onto Mg-Fe-Ti-LDHs decreases in the order $PO_4^{3-} > SO_4^{2-} > NO_3^-$. If an anion is stereochemically more suitable to be added to the interlayer of LDHs, then it has the ability to displace and release the previously present interlayer anion from the LDHs. Therefore, in the desorption experiments, the desorption solution was made by the sodium salt of $PO_4^{3-}$.

As shown in Table 7, the desorption rates of Mg-Fe-Ti-LDHs-$Cl^-$-2 and Mg-Fe-Ti-LDHs-$CO_3^{2-}$ are 49.7% and 37.5%, respectively. Under the same conditions, the desorption rate of Mg-Fe-Ti-LDHs-$Cl^-$-2 on vanadate is significantly higher than that of Mg-Fe-Ti-LDHs-$CO_3^{2-}$. The adsorption experiment showed that the adsorption capacity of Mg-Fe-Ti-LDHs-$Cl^-$-2 on vanadate is higher than the adsorption capacity of Mg-Fe-Ti-LDHs-$CO_3^{2-}$; therefore, it can be seen that the binding force of $CO_3^{2-}$ in Mg-Fe-Ti-LDHs materials is stronger than that of $Cl^-$ in LDHs materials, and that Mg-Fe-Ti-LDHs materials are more inclined to bind $CO_3^{2-}$ ions.

**Table 7.** Desorption experiment of Mg-Fe-Ti-LDHs-$Cl^-$-2 and Mg-Fe-Ti-LDHs-$CO_3^{2-}$.

| LDHs Material | Desorption Solution | $PO_4^{3-}$ Concentration (mg/L) | Desorption Rate (%) |
|---|---|---|---|
| LDHs-$Cl^-$-2 | $Na_3PO_4$ | 500 | 49.7 |
| LDHs-$CO_3^{2-}$ | $Na_3PO_4$ | 500 | 37.5 |

## 4. Conclusions

The novel layered double hydroxides containing carbonate and chloride intercalation were synthesized and then applied to adsorb vanadate. Experimental results indicated

that the material exhibited good vanadate adsorption capacity, and that adding titanium to the LDHs material significantly increases the adsorption capacity of vanadate. The results of isotherm and kinetic experiments indicate that the process of vanadate adsorption by the material is dominated by the occupation of the surface active sites by vanadate. The thermodynamic results indicate that vanadate adsorption on LDHs is an endothermic process. The solution pH has a large effect on the adsorption process, and the adsorption of vanadate by coexisting ions is related to the valence of the anions in the order of $PO_4^{3-} > SO_4^{2-} > NO_3^-$. The main mechanism of vanadate adsorption on LDHs is a ligand exchange process with the OH groups on the layer and an ion exchange process with chloride ions and carbonate ions on the interlayer.

**Author Contributions:** Y.G.: Investigation, Data curation, Visualization, Writing—Original draft preparation, Funding acquisition; Z.Z.: Conceptualization, Validation, Supervision, Reviewing, Project administration, writing—review and editing; H.L.: Visualization, Writing—Original draft preparation, Editing, Funding acquisition; B.H.: Supervision; T.Z.; Project administration. All authors have read and agreed to the published version of the manuscript.

**Funding:** This research was funded by the National Natural Science Foundation of China (Grant No. 41506056) and the Research Project of Shanghai Urban Construction Vocational College (Grant No. CJKY202002).

**Institutional Review Board Statement:** Not applicable.

**Informed Consent Statement:** Not applicable.

**Conflicts of Interest:** The authors declare no conflict of interest.

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
