# Peer review of "Efficient Vanadate Removal by Mg-Fe-Ti Layered Double Hydroxide"

_water, doi:10.3390/w14132090_

Round 1

Reviewer 1 Report

The layered double hydroxides (Mg-Fe-Ti-LDHs) as adsorbents for vanadate are investigated. Adsorption mechanisms, the influence of acidity and the co-ionic mechanism are investigated. The manuscript corresponds to the subject matter of the journal Water and may be submitted to the Section Waste Water Treatment and Reuse.

11)      From the methodology of obtaining samples and conducting experiments it is not clear how coexisting anions (NO3-, SO42-, and PO43-) were introduced

22)      Specify which sodium vanadate was used (sodium metavanadate, sodium orthovanadate, sodium decavanadate) and the purity of the reagent.

33)      What byproducts are obtained for Mg-Fe-Ti-LDH-Cl-5, Fig. 1?

44)      Could the low vanadate adsorption capacity (qm) for LDHs-CO32- compared to other compositions be the reason for the lower adsorbent performance (table 1)?

55)      Literature citations do not satisfy the requirements of the journal:

Author 1, A.B.; Author 2, C.D. Title of the article. Abbreviated Journal Name YearVolume, page range.

Author 1, A.; Author 2, B. Title of the chapter. In Book Title, 2nd ed.; Editor 1, A., Editor 2, B., Eds.; Publisher: Publisher Location, Country, Year; Volume 3, pp. 154–196.

Author Response

Please see the attchment.

Reviewer 2 Report

The manuscript is about synthesing and testing a improved material for cleaning polluted waters. The synthesis and the characterisation has been done in a proper manner although the new material is not compared to a material that is currently use for water purification. So the reader is left to wonder if the material makes any sense. More importantly is that I have serious doubts if the authors have much knowledge about ion adsorption research. The adsorption experiments seem to have been done in a strange manner: without background electrolytes. I hope they have not made this mistake, and that the info is just missing in the manuscript.If the data are just missing, then I advice the authors to write more clearly. Also the adsorption experiments should have been done in a normal manner, using relevant concentrations of vanadium and other ions. If the authors admit that they have done the experiment in a rather unconventional manner, which makes the results difficult to compare with other studies, the reader can accept this.

If the authors have not used background electrolytes, then I strongly advice to do these experiments again with background electrolytes, and not to publish this work. The I would need to advice to reject the manuscript.

Some details.

35 ”has become a major challenge”. Please refer to good and recent literature, for example Gustaffson https://doi.org/10.1016/j.apgeochem.2018.12.027 . Please mention the speciation of vanadium in polluted waters. It is not always vanadate.  See also https://www.rivm.nl/bibliotheek/rapporten/601714021.pdf

47 “they are often used”, I would say, they are often studied. The commercial use is limited I think. Otherwise, you can mention the amount use in China or other countries to make this clear.

49-63. The aspect why vanadate and LDH was chosen is still not very clear. Do other solutions not work for vanadate? I guess the hypothesis is that novel LDH can be made, and are possible better than other LDH’s and other materials.

61 “coexisting ions”. It is good that you include phosphate etc, but you do not mention that in real systems, in waste water treatment, also humic and fulvic acids interfere with the adsorption of oxyanions. For example: Weng, https://doi.org/10.1021/es9000196

108 “pH of the solution was set to 5.0”. How to you “set” pH? Did you adept it after some hours?

Chapter 2.3. What is the ionic strength, or what did you use as background salt? When you do not use a background salt the adsorption data can be interpreted in the wrong way. If no background salt is used all anions are adsorbed to compensate for the positive charge of the LDH: you can not distinguse between chemical adsorption and ion exchange. Normally researchers use a background salt. In that case simple ion exchange is excluded due to the large amount of background ions.

Because you always have some background salt, due to  setting pH, and due to the synthesize of the LDH, there is always salt. So you always use more background salt, because in that case this contamination is not relevant.

You did wash the LDH, but as the LDH is charged, there will be adsorption from some ions used in the synthesize.

The number of replicas is not mentioned although the results show error bars.

120 “1:10 ration”. This is a very strange choice. Normally researchers use a background concentration, and realistic concentrations of relevant ion that can influence the adsorption of the ion that one is interested in. See for example: HCO3 when studying arsenate https://doi.org/10.1021/es063087i, or phosphate when studying vanadate https://doi.org/10.1071/EN16174.

In realistic systems sulphate and bicarbonate a very weakly bound by metalhydroxides, but due to the high concentrations in natural and polluted waters, the effect on specific ions such as vanadate is probably very important. See for example the effect of phosphate on vanadate adsorption on ferryhydrite. https://pub.epsilon.slu.se/14712/1/larsson_et_al_171114.pdf.

111. “1 to 20 mg/l V”. That is a very high concentration. Polymerization cannot be excluded, certainly not on surfaces. This influences you conclusion when you extrapolate to real systems.

Figure 7. “thermodynamic”. What is thermodynamically about an adsorption isotherm?

It is actually not a isotherm is you do not keep all factors constant. The salt concentration and pH is probably not constant.

357. The fact that a little bit of nitrate can desorb 50% of the vanadate means that the adsorption is very weak. You have not compared it to other materials such as goethite or ferrihydrite. So it is not clear if the LDH is usefull.

360-362. The distinction between ion exchange and chemical adsorption has not been made experimentally. The strong effect of nitrate suggest ion exchange. But the research seems not to be done in a proper manner to make it clear.

Author Response

Please the attchment.

Reviewer 3 Report

The authors reported the manuscript entitled "Efficient Vanadate Removal by Mg-Fe-Ti Layered Double Hydroxide" very well. The article is essential for the scientific community but some points must be considered before publication.

1.     Inline no. 96-98 authors used BET and BJH method for surface area analysis, so what are the surface area, pore-volume, and pore size distributions of Mg-Fe-Ti-LDHs materials. Draw its graphs also.

2.     How many times have been reused or recycled the Mg-Fe-Ti-LDHs materials for vanadate removal.

3.     In Mg-Fe-Ti-LDHs materials because of Fe3+ ion, do they also have some magnetic property? How are they separated from the aqueous solution after adsorption process?

4.     How many time intervals were taken to study the time effect parameter? The authors should mention it adsorption experiment section.

 5. On page no. 9, Table 1 in R2 Columns shows some other numbers. Please check this.

6. Please check references.

Reviewer 4 Report

dear authors, thank you for the interesting work. The removal of various heavy metals are important and the use of clay minerals and zeolites are of great importance that might be in line with hydroxides as in the paper

some changes and amendments are needed:

1) the objective of the study and derived tasks are not shown - this should be put at the end of Introductory part

2) Introduction part is too short, can be amended with literature review. Some suggestions: sorption studies by Ozola-Davidane et al; Krauklis et al and similar

3) Figures resolution and illustrative visionary might be improved

4) minor spell check needed

Round 2

Reviewer 1 Report

The manuscript may be published as is

Author Response

Thank you very much for your comment.

Reviewer 2 Report

From the reply of the authors it is clear that they have not used a background salt. Their excuse is: also some other researchers do this. That is really not a good excuse. It is a serious flaw. One should try to give facts. By accepting this, you get a growing amount of papers with results that cannot be compared properly.

I need to explain this. There are always background salt when you synthesize minerals, even after washing with demi. So you can have two choices: you analyses the background salts (this has not been done), or you add extra salts to ensure a certain background solution (the method use mostly)(this has not been done either). The authors do not explain their choice.

This is problematic for all papers that will follow in the future: also from themselves. Because the next time that they try to publish results, with rather different experimental results, cannot be compared to this paper.

If the authors, and also reviewer, are not honest about these "mistakes" then you end up with series of manuscript with the same mistake. It would be helpful if the authors just state something as:

this was not optimal choice, better to add background salts.

Author Response

Response: Thank you very much for your comment.

In the last response to the reviewer, we have answered this comment. For all this, we respect the reviewer’s comment and suggestion, and the discussion on how to do this kind of research may have different opinions from different experts. As the reviewer suggested, it is better to add background salts. Meanwhile, a general research method should be considered as a way.

In this kind of research, as a lot of previous publications in peer reviewed journals, the effect of background salts usually was discussed as coexistent ions or species. In our manuscript, the factors affecting the adsorption process were experimentally evaluated, including solution pH, reaction time, initial pollutant concentration and coexisting ions. Since the real environmental situation is quite complex, the total effect and results are more important to evaluate the performance of the adsorbents to remove the pollutants.

Reviewer 3 Report

The authors have revised the manuscript in light of the referee's report. The paper may be accepted for publication in its current form.

Author Response

Thank you very much for your comment.

Reviewer 4 Report

Minor spell check

literature nr 9. Instead of the author "Kurauklis" should be "Krauklis", please check, looks wrong

Author Response

Thank you very much for your comment. The author’s name has been checked and revised. Please refer to line 430-431 of page 15 in the revised manuscript.

Round 3

Reviewer 2 Report

- I would like it if you include this discussion (one sentence is enough) about background salt. It is relevant for new studies etc.

-One thing is still unclear to me. You use a solution of sodium metavanadate NaVO3. You also write vanadate as VO3 - . But in most polluted water is is probably VO4 3- (V), and probably not a polymer. The adsorption might be higher for a polymer on many materials, but lower on a layered material. I have not seen a remark about this, and about the speciation you expect in solution

Author Response

This manuscript is a resubmission of an earlier submission. The following is a list of the peer review reports and author responses from that submission.